# The impact of race and ethnicity on outcomes in 19,584 adults hospitalized with COVID-19

**Ann M. Navar**[1]*, **Stacey N. Purinton**[2], **Qingjiang Hou**[2], **Robert J. Taylor**[2], **Eric D. Peterson**[1]

**1** University of Texas Southwestern Medical Center, Dallas, TX, United States of America, **2** Cerner Corporation, Kansas City, MO, United States of America

* Ann.navar@utsouthwestern.edu

**Data Availability Statement:** Data from the Cerner Real World Data COVID dataset are available on request to Cerner corporation (contact via Kendra. Stillwell@Cerner.com). Per the user agreements between Cerner and the contributing health

## Abstract

### Introduction

At the population level, Black and Hispanic adults in the United States have increased risk of dying from COVID-19, yet whether race and ethnicity impact on risk of mortality among those hospitalized for COVID-19 is unclear.

### Methods

Retrospective cohort study using data on adults hospitalized with COVID-19 from the electronic health record from 52 health systems across the United States contributing data to *Cerner Real World Data*[TM]. In-hospital mortality was evaluated by race first in unadjusted analysis then sequentially adjusting for demographics and clinical characteristics using logistic regression.

### Results

Through August 2020, 19,584 patients with median age 52 years were hospitalized with COVID-19, including n = 4,215 (21.5%) Black and n = 5,761 (29.4%) Hispanic patients. Relative to white patients, crude mortality was slightly higher in Black adults [22.7% vs 20.8%, unadjusted OR 1.12 (95% CI 1.02–1.22)]. Mortality remained higher among Black adults after adjusting for demographic factors including age, sex, date, region, and insurance status (OR 1.13, 95% CI 1.01–1.27), but not after including comorbidities and body mass index (OR 1.07, 95% CI 0.93–1.23). Compared with non-Hispanic patients, Hispanic patients had lower mortality both in unadjusted and adjusted models [mortality 12.7 vs 25.0%, unadjusted OR 0.44(95% CI 0.40–0.48), fully adjusted OR 0.71 (95% CI 0.59–0.86)].

### Discussion

In this large, multicenter, EHR-based analysis, Black adults hospitalized with COVID-19 had higher observed mortality than white patients due to a higher burden of comorbidities in Black adults. In contrast, Hispanic ethnicity was associated with lower mortality, even in fully adjusted models.

systems, all research requests must be approved by the governance council which consists of representatives from both Cerner and contributing health systems. Upon approval, researchers can access the de-identified datasets on the Cerner platform; individual patient data (even de-identified data) are never released outside of the analytic environment to protect patient confidentiality based on the agreements in place between Cerner and participating sites.

**Funding:** This study was supported by Cerner Corporation who provided access to the dataset and employee time to participate on the research study. The dataset for this study was created by Cerner for use by academic researchers with support from Amazon Web Services, and is available for access to external researchers upon request. Cerner Corporation-employed co-authors (QH, RT, SP) received salary from Cerner corporation during the time they participated on this research project. Cerner Corp. provided in-kind support for the study through their effort and through providing access to data. Cerner Corp. co-authors were a part of the study team and had roles in creating the dataset and data acquisition (RT, SP), data analysis (QH), the concept and design of the study, data interpretation, and critical revision of the manuscript (QH, RT, SP). EP and AMN receive fees for research consulting to Cerner Corporation outside of the present work. EP and AMN were not compensated for their work on this manuscript.

**Competing interests:** I have read the journal's policy and the authors of this manuscript have the following competing interests: Ann Marie Navar and Eric Peterson receive support for consulting to Cerner Corporation for activities outside of this work. Rob Taylor, Qingjiang Hou, and Stacey Purinton are employees of Cerner Corporation. Cerner Corporation-employed co-authors (QH, RT, SP) received salary from Cerner corporation during the time they participated on this research project. Cerner Corp. provided in-kind support for the study through their effort and through providing access to data. Cerner Corp. co-authors were a part of the study team and had roles in creating the dataset and data acquisition (RT, SP), data analysis (QH), the concept and design of the study, data interpretation, and critical revision of the manuscript (QH, RT, SP). EP and AMN receive fees for research consulting to Cerner Corporation outside of this work. EP and AMN were not compensated for their work on this manuscript. This does not alter our adherence to PLOS ONE policies on sharing data and materials.

## Introduction

The COVID-19 pandemic caused approximately 375,000 deaths in the United States in 2020, and was either the cause of death or contributing cause of death for 11.3% of all deaths in the United States [1]. Likely due to COVID-19, the age-adjusted mortality increased in the United States by 15.9% in 2020 [2]. Among patients hospitalized for COVID-19, the mortality rate has generally decreased over time, but has been shown to vary across hospitals, with one study showing a 50% variability in risk-standardized event rates of mortality or referral to hospice among COVID-19 patients in the first six months of the pandemic [3].

COVID-19 has also had a disproportionate impact on Black and Hispanic populations [4, 5]. Specifically, Black and Hispanic adults in the United States are at increased risk of severe infection and are nearly 3 times more likely to die of COVID-19 compared with non-Hispanic white person [6]. It has been less clear, however, whether Black and Hispanic patients hospitalized with COVID-19 have higher mortality. To date, reports on the impact of race and ethnicity on mortality in patients hospitalized with COVID-19 have come from either single center databases, specialized populations such as the Veterans Affairs hospital, or from selected centers participating in a prospective registry [7–11]. Accurate information regarding the impact of race and ethnicity on outcomes is important to help physicians identify higher risk patients when admitted, to identify potential biological mechanisms affecting prognosis, and to target public health interventions appropriately. Most importantly, differences in outcomes by race and ethnicity may serve as an indicator of systemic biases in the healthcare system, with differential treatment leading to differential outcomes.

National data from electronic health records (EHR) from community hospitals offer the ability to evaluate risk factors for complications of COVID-19 and mortality. In 2020, Cerner Corporation, partnered with Amazon Web Services to create a national EHR-based deidentified dataset of all patient hospitalized with COVID-19 who participate in *Cerner Real World Data*$^{TM}$.

Using this dataset, we sought to evaluate race and ethnic differences in patients hospitalized for COVID-19 from across the United States, and the impact of race and ethnicity on mortality.

## Methods

This study was deemed exempt from human subjects review as it used de-identified data by the Duke University Institutional Review Board (Pro00105396). Data from Cerner's COVID-19 database derived from *Cerner Real World Data*$^{TM}$ were used to identify patients hospitalized with COVID-19. The dataset for this analysis was created in August 2020 using data through August 3, 2020. In order to de-identify the data, all dates are shifted up to 8 weeks for each patient, therefore the shifted dates in the study dataset (as opposed to the actual dates of the hospitalizations) ranged from December 1, 2019 through September 2, 2020. For this analysis, we identified all patients age 18 or older who were hospitalized with COVID-19, including those with a positive lab test for SARS-CoV-2 during or within 2 weeks of the hospitalization, as well as patients hospitalized with a diagnosis consistent with COVID-19 disease (S1 Table). Laboratory tests assessed included antigen-based tests; antibody tests were not used. Patients with positive laboratory tests for other coronaviruses than SARS-CoV-2 were also excluded. Only one hospitalization per patient was used; for patients with multiple hospitalizations the most recent hospitalization was considered. For patients admitted from the emergency room,

emergency room visit information was combined with the inpatient admission to create a single episode of care.

Demographic data available on the study sample patients included age, sex, race, insurance status, ethnicity, and first digit of ZIP code. For some patients, race and ethnicity data were reported multiple times across different encounters. To determine race and ethnicity, we evaluated for the presence of an indicator of race within 3 years of and including the analysis hospitalization. For patients with different races listed across different encounters, the patient's race was set to "multiple races." Geographic location for patients included first digit of zip code for the patient. Encounter information for the study COVID-19 hospitalization included admission and discharge dates (shifted within patient to de-identify as above), medications administered, procedures, diagnoses, and laboratory data.

Patient comorbidities at the time of hospital admission, were defined using international classification of disease, version 10 (ICD-10) diagnosis codes, MEDCIN codes, and SNOMED codes recorded during previous inpatient and outpatient encounters in the same healthcare system in the past 3 years. Patients without any encounters in the Cerner electronic health record prior to hospitalization were excluded from models that included comorbidities. Obesity was defined using the body mass index (BMI) or weight and height measurements taken during the hospital visit, or if not available, then the most recent body mass index measurement prior to the index admission was used, with a cutoff for obesity of $> 30.0$ kg/m$^2$.

Clinical complications of COVID-19 were identified using diagnosis and procedure codes from the hospitalization. Mortality was assessed among patients either discharged alive or who died during the index stay. Patients transferred to other facilities, those discharged to hospice, and those still admitted at the time of dataset creation were excluded from the analysis.

Descriptive statistics are presented for characteristics and outcomes for patients hospitalized COVID-19 by race and ethnicity, with t-tests used for continuous variables and Pearson Chi-squared tests or Fisher's exact tests used for categorical variables when applicable.

To evaluate the impact of race and ethnicity on the risk of mortality, univariable logistic regression was first used to evaluate the association between sex, race, ethnicity, insurance, geographic region, age, and calendar date of admission. Age and date of admission were modeled using a restricted cubic spline (RCS) function to account for their non-linear effect with knots selected at 5 (5, 27.5, 50, 72.5, 95) and 3 (27.5, 50, 72.5) percentile points, and then considered as such for inclusion in the multivariable model, respectively. Multivariable logistic regression was then performed to evaluate the association between nonclinical factors and the in-hospital mortality including age, sex, race, ethnicity, insurance category, geographic region, and time of hospital admission. Patients with sex as either missing or "other" were excluded from the multivariable analysis given small overall numbers (n = 47). Patients with missing data for race, ethnicity, insurance status, or ZIP code were not excluded; these variables were modeled as "missing". Reference categories for race and ethnicity were white and non-Hispanic, respectively.

Next, a multivariable model was created to evaluate the association between race and ethnicity and outcomes further adjusting for patient comorbidities including asthma, coronary artery disease, chronic kidney disease, chronic obstructive pulmonary disease, diabetes, end stage renal disease, heart failure, hypertension, and BMI. This analysis only included those who had at least one prior visit in the dataset to determine prior comorbidities and those for whom BMI data were available. BMI was modeled with RCS function with knots selected at 3 (27.5, 50, 72.5) percentile points to account for nonlinearity. All demographic, race, and ethnicity variables were included in the model, with stepwise variable selection used for clinical variables with a retention p-value of <0.05.

## Results

### Cohort characteristics and unadjusted outcomes

We identified 28,299 patients hospitalized with COVID-19 during the analytic window. At the time the dataset was created, n = 1729 (6.1%) were still hospitalized, n = 466 (1.6%) had been discharged to hospice, n = 4689 (16.6%) were transferred to another facility, and discharge status was unknown for n = 1,831 (6.5%). The breakdown of those with unknown discharge status is presented in S2 Table; Black patients had the highest rates of unknown discharge status while white patients had highest rates of transfers. This left a total of 19,584 patients for whom discharge disposition was available. Race data was missing for n = 1,017 (5.19%) of patients overall.

### Differences in hospitalized cases by race and ethnicity

Overall, 51.0% (n = 9994) of our sample was white while 21.5% were recorded as Black (n = 4215). Table 1 shows differences between Black and white adults admitted with COVID-19. Black adults were younger (median age 59 vs 62 years), less likely to be male (47.3% vs 53.0%), less likely to be Hispanic (1.9% vs 36.3%), had higher BMIs (median BMI 30.7 vs 38.9), and different distribution of insurance coverage (p<0.001 for all). Black patients also had higher rates of diabetes, hypertension, coronary artery disease, heart failure, chronic kidney disease, and end stage renal disease, and lower rates of COPD and asthma (see Table 1).

Table 1 also shows characteristic of adults stratified by ethnicity. Among those for whom ethnicity data were available (n = 17,030), n = 5,761 were Hispanic (33.8%) while 11,269 were non-Hispanic. Compared with non-Hispanic adults, Hispanic patients were older, more likely to be male, and less likely to be non-white race. Statistically significant differences were also seen in the geographic distribution of patients by ethnicity as well as the insurance type. Rates of comorbidities including diabetes, hypertension, heart failure, COPD, asthma, and CAD were all lower in Hispanic adults, while ESRD and was slightly higher.

### Factors associated with mortality

Of the 19,584 patients included, n = 4050 (20.7%) died during the hospital stay. Table 2 shows characteristics of adults overall and stratified by in-hospital mortality. Among those that died, the median length of stay was 7.9 days (interquartile range 3.6–14.5 days). Among those who were discharged alive, median length of stay was 4.5 days (IQR 2.4–8.1 days). Complication rates were low overall: 4.96% of patients had a myocardial infarction, 1.47% had stroke, 1.80% ventricular tachycardia, and 2.03% pulmonary embolism.

Statistically significant differences (p<0.05) were seen in survival across a number of demographic and clinical variables with increasing mortality seen in older adults, males, those with comorbidities, by ZIP code, insurance status, and BMI. Age and BMI were nonlinear in their association with mortality and were modeled using restricted cubic splines (S1 and S2 Figs). Time of admission was also associated with changes in the risk of mortality in a nonlinear fashion with increasing mortality early, peaking in mortality around May, and then decreased through the first three quarters of 2020 (S3 Fig).

### Mortality and complications by race and ethnicity

Table 3 shows rates of mortality and complications among adults hospitalized with COVID-19 by race and ethnicity. Among the n = 4215 Black adults, 955 (22.7%) died, whereas n = 2078 of 9994 (20.8%) white adults died, a difference that was statistically significant in univariable analysis (OR 1.12, 95% CI 01.02–1.22, p = 0.013).

**Table 1. Characteristics of white, Black, Hispanic, and non-Hispanic patients hospitalized with COVID-19.**

| | White N = 9994 | Black N = 4215 | Hispanic N = 5761 | Non-Hispanic N = 11,269 |
|---|---|---|---|---|
| Age | | | | |
| Median (IQR) | 62 (49, 76) | 59 (47, 71) | 62 (49, 76) | 59 (47, 71) |
| 18–39 | 1,432 (14.33) | 671 (15.92) | 1,198 (20.8) | 1,499 (13.3) |
| 40–49 | 1,137 (11.38) | 556 (13.19) | 937 (16.26) | 1,227 (10.89) |
| 50–59 | 1,846 (18.47) | 903 (21.42) | 1,236 (21.45) | 2,126 (18.87) |
| 60–69 | 1,974 (19.75) | 919 (21.8) | 1,049 (18.21) | 2,430 (21.56) |
| 70–79 | 1,737 (17.38) | 710 (16.84) | 684 (11.87) | 2,054 (18.23) |
| 80–89 | 1,863 (18.64) | 453 (10.75) | 652 (11.32) | 1,927 (17.10) |
| ≥90 | 5 (0.05) | 3 (0.07) | 5 (0.09) | 6 (0.05) |
| Sex | | | | |
| Male | 5,297 (53.00) | 1,994 (47.31) | 3,045 (52.86) | 5,758 (51.10) |
| Race | | | | |
| American Indian or Alaska | | | 23 (0.40) | 438 (3.89) |
| Asian or Pacific islander | | | 24 (0.42) | 626 (5.56) |
| Black or African American | | | 80 (1.39) | 3,955 (35.10) |
| Mixed racial group | | | 1 (0.02) | 3 (0.03) |
| Other racial group | | | 1,501 (26.05) | 426 (3.78) |
| Unknown racial group | | | 500 (8.68) | 228 (2.02) |
| White | | | 3,632 (63.04) | 5,593 (49.63) |
| Ethnicity | | | | |
| Hispanic | 3,632 (36.34) | 80 (1.90) | | |
| BMI | | | | |
| Median (IQR) | 28.9 (25.0, 34.1) | 30.7 (26.0, 37.0) | 29.3 (25.9, 34.0) | 29.2 (24.9, 35.0) |
| <25 | 1,816 (25.49) | 778 (20.36) | 1,281 (33.62) | 2,781 (28.49) |
| 25–29.9 | 2,202 (30.91) | 983 (25.73) | 753 (19.76) | 2,509 (25.70) |
| ≥30 | 3,106 (43.6) | 2,060 (53.91) | 1,776 (46.61) | 4,471 (45.80) |
| ZIP (first digit)* | | | | |
| 0 | 1,299 (14.97) | 670 (16.55) | 501 (10.06) | 1,766 (16.25) |
| 1 | 751 (8.66) | 206 (5.09) | 217 (4.36) | 995 (9.15) |
| 2 | 634 (7.31) | 1,841 (45.47) | 100 (2.01) | 2,524 (23.22) |
| 3 | 1,654 (19.07) | 481 (11.88) | 1,289 (25.87) | 941 (8.66) |
| 4 | 829 (9.56) | 314 (7.76) | 143 (2.87) | 1,076 (9.90) |
| 5 | 93 (1.07) | 15 (0.37) | 22 (0.44) | 90 (0.83) |
| 6 | 321 (3.7) | 82 (2.03) | 69 (1.38) | 411 (3.78) |
| 7 | 797 (9.19) | 158 (3.90) | 571 (11.46) | 484 (4.45) |
| 8 | 380 (4.38) | 121 (2.99) | 244 (4.9) | 1,054 (9.70) |
| 9 | 1,917 (22.10) | 161 (3.98) | 1,826 (36.65) | 1,528 (14.06) |
| Insurance | | | | |
| Uninsured | 587 (5.87) | 151 (3.58) | 557 (9.67) | 333 (2.96) |
| Medicare | 3,098 (31.00) | 1,442 (34.21) | 807 (14.01) | 4,187 (37.16) |
| Medicaid | 1,042 (10.43) | 704 (16.70) | 1,022 (17.74) | 1,448 (12.85) |
| Government | 166 (1.66) | 78 (1.85) | 70 (1.22) | 241 (2.14) |
| Private | 3,185 (31.87) | 1,232 (29.23) | 1,710 (29.68) | 3,274 (29.05) |
| Other | 239 (2.39) | 50 (1.19) | 124 (2.15) | 267 (2.370) |
| Missing | 1,677 (16.78) | 558 (13.24) | 1,471 (25.53) | 1,519 (13.48) |
| Comorbidities | | | | |
| N (%) with comorbidity data available | 7,437 (74.41) | 3,228 (76.58) | 3,857 (66.95) | 8,582 (76.16) |

*(Continued)*

**Table 1.** (Continued)

|  | White N = 9994 | Black N = 4215 | Hispanic N = 5761 | Non-Hispanic N = 11,269 |
|---|---|---|---|---|
| Diabetes | 2,114 (28.43) | 1,205 (37.33) | 1,144 (29.66) | 2,783 (32.43) |
| Hypertension | 3,692 (49.64) | 1,940 (60.10) | 1,645 (42.65) | 4,788 (55.79) |
| Heart Failure | 1,110 (14.93) | 545 (16.88) | 398 (10.32) | 1,452 (16.92) |
| ESRD | 273 (3.67) | 313 (9.70) | 202 (5.24) | 500 (5.83) |
| COPD | 1,036 (13.93) | 357 (11.06) | 253 (6.56) | 1,231 (14.34) |
| Asthma | 701 (9.43) | 414 (12.83) | 318 (8.24) | 920 (10.72) |
| Coronary artery disease | 1,517 (20.40) | 576 (17.84) | 528 (13.69) | 1,793 (20.89) |

*ZIP: ZIP code data exclude those with missing ZIP code

*p-value for differences by race and ethnicity all <0.001 with two exceptions: difference in coronary artery disease by race p-value was 0.03, and diabetes prevalence difference by ethnicity p-value was 0.002.

IQR = interquartile range, ESRD = end stage renal disease, COPD = chronic obstructive pulmonary disease

Non-Hispanic adults had higher mortality rates than Hispanic adults (12.7% vs 25.0%, OR 0.44, 95% CI 0.40–0.48, p<0.001). In addition to mortality, Hispanic adults had lower rates of complications of COVID-19 including lower rates of myocardial infarction, stroke, ventricular tachycardia, and pulmonary embolism (p<0.001 for all). Compared with White adults, Nlack adults had similar rates of myocardial infarction, but higher rates of stroke, ventricular tachycardia, and pulmonary embolism.

In multivariable modeling adjusting for demographic factors (age, sex, race, ethnicity, ZIP code and time of admission), Black race was associated with increasing risk of mortality (OR 1.13, 95% CI 1.01–1.27 compared with white race), and Hispanic ethnicity was associated with lower risk of mortality (OR 0.74, 95% CI 0.65–0.83).

Adjusting for demographic factors, race, and ethnicity, BMI, asthma, diabetes, heart failure, and chronic kidney disease were also associated with the risk of mortality (S3 Table). However, after comorbidities and BMI were included in the model, racial differences were attenuated and no longer statistically significant (Fig 1, OR for Black vs white: 1.07, 95% CI 0.93–1.23), while differences by ethnicity remained statistically significant (OR for Hispanic vs non-Hispanic 0.71, 95% CI 0.59–0.86).

## Discussion

In a nationwide, EHR-based database of 19,584 patients from 52 health systems across the United States, mortality among patients hospitalized for COVID-19 was higher in Black patients compared with white patients. This increasing risk of mortality remained statistically significant after adjusting for demographics such as age and sex. However, Black patients with COVID-19 had a higher burden of comorbid illnesses. As a result, after adjusting for comorbidities racial differences in mortality were no longer statistically significant. In contrast, Hispanic adults had lower overall mortality than non-Hispanic adults, a finding that remained statistically significant even after accounting for demographic and clinical differences among those hospitalized.

Prior findings regarding the association between race, ethnicity, and mortality after hospitalization are mixed. In one single-center study from an academic Medical Center in New York, Black and Hispanic patients hospitalized with COVID-19 had lower mortality rates overall, which remained statistically significant even after adjusting for age, sex, socioeconomic status, and comorbidities [12]. Other studies from either single centers or regions or specialized populations found either lower rates of mortality in Hispanic and Black adults or no

**Table 2. Characteristics of adults hospitalized with COVID-19 overall and stratified by in-hospital mortality.**

|  | Overall | In-Hospital Mortality | Discharged Alive |
|---|---|---|---|
| Overall sample | 19,584 | 4,050 (20.68) | 15,534 (79.32) |
| Length of stay |  |  |  |
| Median (IQR) | 4.97 (2.55, 9.15) | 7.89 (3.60, 14.47) | 4.54 (2.38, 8.05) |
| Age |  |  |  |
| Median (IQR) | 52 (37, 65) | 75 (64, 84) | 49 (35, 61) |
| 18–39 | 3,272 (16.71) | 89 (2.02) | 3,183 (20.49) |
| 40–49 | 2,597 (13.26) | 132 (3.26) | 2,465 (15.87) |
| 50–59 | 3,897 (19.9) | 421 (10.40) | 3,476 (22.38) |
| 60–69 | 3,974 (20.29) | 820 (20.25) | 3,154 (20.30) |
| 70–79 | 3,058 (15.61) | 1,084 (26.77) | 1,974 (12.71) |
| 80–89 | 2,775 (14.17) | 1,498 (36.99) | 1,277 (8.22) |
| ≥90 | 11 (0.06) | 6 (0.15) | 5 (0.03) |
| Sex |  |  |  |
| Female | 9,294 (47.46) | 1,689 (41.7) | 7,605 (48.96) |
| Male | 10,243 (52.3) | 2,353 (58.1) | 7,890 (50.79) |
| Other/missing | 47 (0.24) | 8 (0.20) | 39 (0.24) |
| Race |  |  |  |
| American Indian or Alaska | 474 (2.42) | 114 (2.81) | 360 (2.32) |
| Asian or Pacific islander | 679 (3.47) | 134 (3.31) | 545 (3.51) |
| Black or African American | 4,215 (21.52) | 955 (23.58) | 3,260 (20.99) |
| Mixed racial group | 4 (0.02) | 1 (0.02) | 3 (0.02) |
| Other racial group | 3,201 (16.34) | 497 (12.27) | 2,704 (17.41) |
| Unknown racial group | 1,017 (5.19) | 271 (6.69) | 746 (4.8) |
| White | 9,994 (51.03) | 2,078 (51.31) | 7,916 (50.96) |
| Ethnicity |  |  |  |
| Ethnic group unknown | 2,554 (13.04) | 503 (12.42) | 2,051 (13.20) |
| Hispanic or Latino | 5,761 (29.42) | 732 (18.07) | 5,029 (32.37) |
| Not Hispanic or Latino | 11,269 (57.54) | 2,815 (69.51) | 8,454 (54.42) |
| BMI |  |  |  |
| Mean (STD) | 30.46 (7.31) | 29.11 (7.67) | 30.66 (7.23) |
| <25 | 3,624 (18.5) | 1,101 (27.19) | 2,523 (16.24) |
| 25–29.9 | 4,801 (24.51) | 1,037 (25.6) | 3,764 (24.23) |
| ≥30 | 7,118 (36.35) | 1,275 (31.48) | 5,843 (37.61) |
| N missing | 4,041 (20.63) | 637 (15.73) | 3,404 (21.91) |
| ZIP |  |  |  |
| 0 | 3,094 (15.8) | 964 (23.8) | 2,130 (13.71) |
| 1 | 1,608 (8.21) | 466 (11.51) | 1,142 (7.35) |
| 2 | 3,260 (16.65) | 733 (18.1) | 2,527 (16.27) |
| 3 | 2,263 (11.56) | 273 (6.74) | 1,990 (12.81) |
| 4 | 1,256 (6.41) | 211 (5.21) | 1,045 (6.73) |
| 5 | 120 (0.61) | 8 (0.20) | 112 (0.72) |
| 6 | 499 (2.55) | 85 (2.10) | 414 (2.67) |
| 7 | 1,103 (5.63) | 244 (6.02) | 859 (5.53) |
| 8 | 1,353 (6.91) | 288 (7.11) | 1,065 (6.86) |
| 9 | 3,441 (17.57) | 609 (15.04) | 2,832 (18.23) |
| **Insurance** |  |  |  |
| Uninsured | 1,057 (5.4) | 76 (1.88) | 981 (6.32) |

*(Continued)*

**Table 2.** (Continued)

|  | Overall | In-Hospital Mortality | Discharged Alive |
|---|---|---|---|
| Medicare | 5,459 (27.87) | 2,138 (52.79) | 3,321 (21.38) |
| Medicaid | 2,939 (15.01) | 304 (7.51) | 2,635 (16.96) |
| Government | 322 (1.64) | 45 (1.11) | 277 (1.78) |
| Private | 5,945 (30.36) | 657 (16.22) | 5,288 (34.04) |
| Other | 418 (2.13) | 163 (4.02) | 255 (1.64) |
| Missing | 3,444 (17.59) | 667 (16.47) | 2,777 (17.88) |
| **Complications**\* |  |  |  |
| Myocardial Infarction | 959 (4.96) | 532 (13.33) | 427 (2.78) |
| Stroke | 284 (1.47) | 159 (3.98) | 125 (0.81) |
| Ventricular tachycardia | 349 (1.80) | 202 (5.06) | 147 (0.96) |
| Pulmonary embolism | 393 (2.03) | 104 (2.61) | 289 (1.88) |
| **Total Length of Stay (days)** | 4.99 (2.57, 9.17) | 7.88 (3.59, 14.47) | 4.56 (2.40, 8.06) |

\*Complications data excludes n = 241 overall (59 who died and 182 who survived) who are missing diagnoses data from hospitalization to evaluate complication rates

\*\*All p-values <0.0001 comparing in-hospital mortality vs discharged alive

IQR = interquartile range, STD = standard deviation, ESRD = end stage renal disease, COPD = chronic obstructive pulmonary disease BMI = body mass index

association in multivariable adjusted analyses [4–9, 13]. Recent data from a national prospective registry found that Black and Hispanic adults represented a disproportionate number of those hospitalized with COVID-19 compared with the general population. In unadjusted analyses, both Hispanic adults and Black adults had lower all-cause mortality compared with non-Hispanic white adults, though these differences were not statistically significant after adjusting for age, medical history, and sociodemographic factors [7].

The findings of this multi-center study with larger patient numbers are slightly different from these prior reports, as Black race did appear to be a risk factor for mortality in unadjusted models and in models adjusting for demographics. However, consistent with other findings,

**Table 3. Mortality and complications among Black, white, Hispanic, and non-Hispanic adults hospitalized for COVID-19.**

|  | Black N = 4215 | White N = 9994 | OR (Black vs white) | p-value | Hispanic N = 5761 | Non-Hispanic N = 11,269 | OR | p-value |
|---|---|---|---|---|---|---|---|---|
| **Mortality** | 955 (22.66) | 2078 (20.79) | 1.11 (1.02–1.22) | 0.014 | 732 (12.71%) | 2815 (24.98%) | 0.44 (0.40–0.48) | <0.001 |
| **Myocardial Infarction** | 214 (5.17) | 487 (4.91) | 1.04 (0.89–1.23) | 0.53 | 234 (4.09) | 603 (5.45) | 0.75 (0.64–0.87) | <0.001 |
| **Stroke** | 76 (1.83) | 137 (1.38) | 1.32 (1.00–1.75) | 0.045 | 47 (0.82) | 185 (1.67) | 0.49 (0.36–0.68) | <0.001 |
| **Pulmonary Embolism** | 105 (2.53) | 174 (1.76) | 1.44 (1.13–1.84) | 0.002 | 78 (1.36) | 263 (2.38) | 0.57 (0.45–0.74) | <0.001 |
| **Ventricular Tachycardia** | 116 (2.80) | 194 (1.96) | 1.43 (1.13–1.80) | 0.003 | 65 (1.14) | 243 (2.20) | 0.52 (0.39–0.68) | <0.001 |
| **Length of Stay Among those Discharged alive** | 4.65 (2.49, 8.10) | 4.25 (2.24, 7.68) |  | 0.024 | 4.37 (2.25, 7.92) | 4.55 (2.42, 8.01) |  | 0.99 |
| **LOS Among deceased** | 7.94 (3.98, 13.87) | 7.02 (3.11, 13.49) |  | 0.75 | 9.07 (4.21, 17.94) | 7.21 (3.29, 13.20) |  | <0.0001 |

+ 69 white and 68 black adults were missing diagnoses codes from hospital stay to determine complications and excluded from the denominator for these calculations, leaving 5724 Hispanic, 11073 non-Hispanic, 5733 white, and 3151 Black adults for the denominator for complications

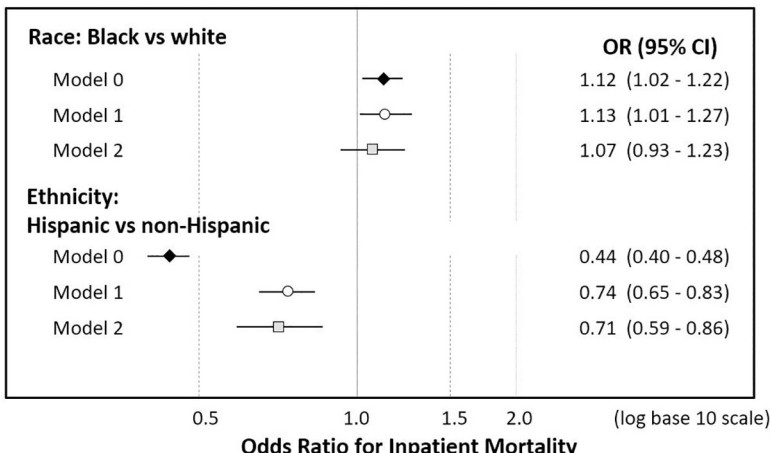

**Fig 1. Association between Black race and Hispanic ethnicity and inpatient mortality.** The figure shows odds ratios and 95% confidence intervals (CI) for race (Black vs white) and ethnicity (Hispanic vs non-Hispanic) in unadjusted models (Model 0, black diamond), adjusting for demographic characteristics (Model 1, white circle), and comorbidities and body mass index (model 2, grey square).

once comorbidities were included the association between Black race and increasing mortality was no longer statistically significant.

This finding should be interpreted with caution. That adjusting for comorbidities attenuated the association between race and mortality does not indicate that there are no differences in mortality from COVID-19 by race particularly given the disproportionate burden of comorbidities associated with increased risk of complications and mortality from COVID-19 such as obesity, hypertension, diabetes, and chronic kidney disease in Black adults in the United States [14]. Rather, our findings suggest that a significant proportion of the difference in mortality between Black and white adults hospitalized with COVID-19 can be attributed to differences in the underlying burden of these comorbidities.

In contrast to race, we did find differences in survival among patients hospitalized with COVID-19 by ethnicity, with Hispanic ethnicity being associated with lower mortality. Compared with non-Hispanic patients, Hispanic patients were older and slightly more likely to be male, but had lower rates of most comorbidities including diabetes, hypertension, and coronary artery disease. In both univariable and multivariable analysis adjusting for these differences, Hispanic ethnicity remained associated with lower mortality. This has been shown in a multi-center study in the past: in a large registry, Hispanic adults had lower mortality compared with non-Hispanic adults, though this difference did not remain statistically significant after multivariable adjustment [7].

On the relative scale, the increased mortality seen in Black compared with white adults hospitalized with COVID-19 is still far less than the relative differences in mortality seen in the overall population [3]. However, compared with their overall representation in the general population, Black and Hispanic adults represented a much greater proportion of patients hospitalized with COVID-19 in our dataset. Specifically, nationwide, 13% of persons in the United States are Black, whereas 22% of patients in our dataset were Black. Similarly, Hispanic persons make up 19% of the overall population, far less than the 29% of patients hospitalized with COVID-19 in our dataset [15]. Thus, we conclude that the disproportionate burden of mortality in Black and Hispanic communities most likely represents their increased prevalence of COVID-19 infections rather than differences in outcome among those hospitalized [1, 2].

Our study also demonstrates the sobering severity of COVID-19. Overall, over 1 in 5 adults hospitalized with COVID-19 died during their inpatient stay, including over 50% of those hospitalized age 80 and older, highlighting the critical need to increase immunization efforts to curb the pandemic. Black adults are less likely to report willingness to be vaccinated, largely due to concern about side effects and decreased reported trust in vaccines [16]. Efforts are needed to improve educational efforts about vaccine safety and efficacy in high-risk communities with a focus on Black communities in order to maximize the impact of vaccination in reducing health disparities [13]. In addition to education to improve vaccine acceptance, vaccine delivery systems should focus on highest risk communities. However, early data raise the concern that Black and Hispanic persons are receiving a lower proportion of vaccines compared with white persons, potentially due to differential access to vaccine [17]. Unfortunately, efforts to prioritize Black and Hispanic at-risk communities have been prohibited in at least one large metropolitan area [18].

This study has several important limitations. First, race and ethnicity were based on EHR data, which is usually captured by patient self-report, but is subject to potential errors in both data entry and incomplete data capture. Our data on race and ethnicity reflect these as social constructs and not biological ones which would have required genetic ancestry. Other limitations of using data from the electronic medical record include incomplete capture of patient comorbidities particularly among those who did not receive care at that institution previously. Excluding those without prior comorbidity data did not appear, however, to impact our mortality estimates: The overall mortality in the sample was 21%, compared with 20% among those who had prior comorbidity data available. Finally, we defined COVID-19 illness using either a positive laboratory test or a clinical diagnosis of COVID-19. This may have included some cases that were misdiagnosed clinically without lab confirmation, or patients hospitalized for other reasons who were incidentally determined to be infected with COVID-19 based on screening lab tests. Finally, discharge status was unavailable in 6.5% of patients, and 16.6% were transferred to another facility, so their ultimate outcomes were unknown. If transfers were variable by race, and survival was different among those who were transferred vs remained hospitalized, this may have impacted our findings, which may only apply to those patients who remain hospitalized.

## Conclusion

Black adults hospitalized for COVID-19 had higher in-hospital mortality than white adults due to increased prevalence of comorbidities that increase the risk of death among adults hospitalized with COVID-19. In contrast, Hispanic adults had lower mortality rates even after accounting for differences in patients hospitalized with disease.

## Supporting information

**S1 Fig.**
(JPG)

**S2 Fig.**
(JPG)

**S3 Fig.**
(JPG)

**S1 Table.**
(PDF)

**S2 Table.**
(PDF)

**S3 Table.**
(PDF)

## Author Contributions

**Conceptualization:** Ann M. Navar, Stacey N. Purinton, Eric D. Peterson.

**Data curation:** Stacey N. Purinton, Robert J. Taylor.

**Formal analysis:** Qingjiang Hou, Robert J. Taylor.

**Investigation:** Ann M. Navar, Robert J. Taylor, Eric D. Peterson.

**Methodology:** Ann M. Navar, Qingjiang Hou, Eric D. Peterson.

**Project administration:** Ann M. Navar.

**Supervision:** Ann M. Navar, Eric D. Peterson.

**Writing – original draft:** Ann M. Navar, Eric D. Peterson.

**Writing – review & editing:** Ann M. Navar, Stacey N. Purinton, Qingjiang Hou, Robert J. Taylor, Eric D. Peterson.

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
