## [Decision Letter · Decision Letter 0]

22 Apr 2021

PONE-D-21-07038

The Impact of Race and Ethnicity on Outcomes in 19,584 adults Hospitalized with COVID-19

PLOS ONE

Dear Dr. Navar,

Thank you for submitting your manuscript to PLOS ONE. After careful consideration, we feel that it has merit but does not fully meet PLOS ONE’s publication criteria as it currently stands. Therefore, we invite you to submit a revised version of the manuscript that addresses the points raised during the review process.

We look forward to receiving your revised manuscript.

Kind regards,

Francesco Di Gennaro

Academic Editor

PLOS ONE

Additional Editor Comments:

Dear Authors follow reviewer suggestion to improve your paper.

Journal Requirements:

2. Thank you for providing the following Funding Statement: 

"This study was supported by Cerner Corporation who provided access to the dataset and employee time to participate on the research study. The dataset for this study was created by Cerner for use by academic researchers with support from Amazon Web Services, and is available for access to external researchers upon request. Ann Marie Navar and Eric Peterson receive support for consulting to Cerner Corporation for activities outside of this work, but were not compensated for work on this analysis. Rob Taylor, Qingjiang Hou, and Stacey Purinton are employees of Cerner Corporation. They were not compensated for their work on this study beyond their regular salary for employment with Cerner."

We note that one or more of the authors is affiliated with the funding organization, indicating the funder may have had some role in the design, data collection, analysis or preparation of your manuscript for publication; in other words, the funder played an indirect role through the participation of the co-authors.

If the funding organization did not play a role in the study design, data collection and analysis, decision to publish, or preparation of the manuscript and only provided financial support in the form of authors' salaries and/or research materials, please review your statements relating to the author contributions, and ensure you have specifically and accurately indicated the role(s) that these authors had in your study in the Author Contributions section of the online submission form. Please make any necessary amendments directly within this section of the online submission form.  Please also update your Funding Statement to include the following statement: “The funder provided support in the form of salaries for authors [insert relevant initials], but did not have any additional role in the study design, data collection and analysis, decision to publish, or preparation of the manuscript. The specific roles of these authors are articulated in the ‘author contributions’ section.”

If the funding organization did have an additional role, please state and explain that role within your Funding Statement.

Please also provide an updated Competing Interests Statement declaring this commercial affiliation along with any other relevant declarations relating to employment, consultancy, patents, products in development, or marketed products, etc.  

Reviewers' comments:

Reviewer's Responses to Questions

**Comments to the Author**

1. Is the manuscript technically sound, and do the data support the conclusions?

Reviewer #1: Yes

Reviewer #2: Yes

2. Has the statistical analysis been performed appropriately and rigorously? 

Reviewer #1: Yes

Reviewer #2: I Don't Know

3. Have the authors made all data underlying the findings in their manuscript fully available?

Reviewer #1: No

Reviewer #2: Yes

4. Is the manuscript presented in an intelligible fashion and written in standard English?

Reviewer #1: Yes

Reviewer #2: Yes

5. Review Comments to the Author

Reviewer #1: Reviewer comment for PLOS ONE

Date of review: 14 April 2021

Manuscript Number: PONE-D-21-07038

Article Type: Research Article

Full Title: The Impact of Race and Ethnicity on Outcomes in 19,584 adults Hospitalized with COVID-19

Suggested title: The Impact of Race and Ethnicity on Outcomes in 19,584 adults Hospitalized with COVID-19 in the United States

Corresponding Author: Ann Marie Navar, UT Southwestern: The University of Texas Southwestern Medical Center, Dallas, UNITED STATES

Reviewer comment

Summary of the research and overall impression

This manuscript identified whether race and ethnicity impact on risk of death among those hospitalized for COVID-19 in the US. Generally, the paper presented well. The introduction is short and required to add more data about the burden of the disease and background situation in the hospitals or States included in the study. Methods session presents how the analysis was carried out. Results are comprehensive, however, a few editions are suggested.

BTW, I don’t see the line number in the manuscript in PDF version and it is difficult to provide the direction of the comments.

Discussion on specific area of improvement

Abstract

Well-written and clearly presented abstract.

1. Results - line 3: “[22.7% vs 20.8%, unadjusted 1.12 (95% CI 1.02-. I think it should be unadjusted OR 1.12.

2. The paper was said “in adult”, please add the mean (or median) age of participants in the results.

Introduction

3. Introduction is too short; it is incomplete information. Suggest to add more data on COVID-19 pandemic outbreak prevalence of (burden) across the US, differences in States, proportion of hospitalized patient and mortality rates, etc. (burden of disease)

4. Suggest to include other studies findings on race and ethnicity diversion on disease burden and mortality in other infectious diseases and any literature on COVID-19 in other countries. No data is the data to present. (gap in knowledge)

5. Suggest to add how race and ethnicity can influence for diagnosis and treatment for preventing life-threatening situation or reduce the mortality in certain diseases, more appropriately with infectious diseases or COVID-19. (gap in knowledge)

6. Lastly, why this study is important to conduct (justification) and how this study can provide information or input to the health system.

Materials and methods

7. Pg. 10, 3rd paragraph, 1st line: “ICD-10 diagnosis codes” – please spell this out and add reference.

8. Pg. 10, 3rd paragraph, 3rd line: “Cerner EHR” – please spell this out. Add reference for using cut off point for obesity.

Results

9. Pg. 12, 1st paragraph: to add text about age (said “younger”, but please specify), sex and geographic distribution of cases

10. Pg. 12, the title is not matched with the text, it is talking about “demographic characteristics and comorbidity of the cases”, not “race and ethnicity”

11. Again, the paragraph does not refer to any table. Please add reference table for the data presented in the text.

12. Tables: the data (numbers) is suggested to right-aligned, rather than centered.

13. Pg. 18, table 3: Please add odds ratios (OR) and CI in the tables. You may use * for p-values that are significant (eg. *p<0.001, #p<0.01)

Discussion and conclusions

14. Pg. 19, 1st paragraph, line 1: please use the exact sample size, do not use “nearly 20,000 patients” and use “52 health systems”.

15. Pg. 20, 2nd paragraph: line 1-3: please add reference.

Recommendation

I would like to propose the minor revision for this manuscript.

Please add the ethical approval number from Duke University.

Reviewer #2: The investigators conducted a retrospective cohort study to evaluate, among patients hospitalized for COVID-19, the differences in race and ethnicity and their association with death.

The manuscript is presented clearly and concisely. The following comments may help in improving the manuscript.

General

1. Persons/patients, consider using one uniformly.

2. Mortality/death consider using one uniformly.

3. clinical comorbidities/comorbidities consider using one uniformly.

Methods

1. “..we utilized any available encounter within 3 years of and including the analysis hospitalization”. Consider revising this text for clarity.

2. “those discharged on hospice”. Should it be“to” instead of “on”?

Results

1. Consider reporting, % of participants for whom data on race and ethnicity were missing.

2. “Compared with White adults, black adults had similar rates of myocardial infarction, but higher rates of myocardial infarction, stroke, ventricular tachycardia, and pulmonary embolism.” In this sentence, should the text “but higher rates of myocardial infarction” be omitted?

Tables

1. Expand abbreviations in footnotes (e.g. ESRD)

2. There is no data in Table 1 on complications, and survival, consider revising the Table title

Discussion

1. Outcome (mortality) data for a significant proportion of participants (about 31%) was not available. Consider mentioning this as a limitation and its likely implications.

Conclusion

1. Consider revising to exclude redundant text and for clarity.

6. PLOS authors have the option to publish the peer review history of their article (what does this mean?). If published, this will include your full peer review and any attached files.

Reviewer #1: **Yes: **Poe Poe Aung

Reviewer #2: No

---

## [Author Response · Author response to Decision Letter 0]

10 Jun 2021

Journal Requirements:

RESPONSE: The manuscript has been reformatted as requested to align with guidelines. 

2. Thank you for providing the following Funding Statement: 

"This study was supported by Cerner Corporation who provided access to the dataset and employee time to participate on the research study. The dataset for this study was created by Cerner for use by academic researchers with support from Amazon Web Services, and is available for access to external researchers upon request. Ann Marie Navar and Eric Peterson receive support for consulting to Cerner Corporation for activities outside of this work, but were not compensated for work on this analysis. Rob Taylor, Qingjiang Hou, and Stacey Purinton are employees of Cerner Corporation. They were not compensated for their work on this study beyond their regular salary for employment with Cerner."

We note that one or more of the authors is affiliated with the funding organization, indicating the funder may have had some role in the design, data collection, analysis or preparation of your manuscript for publication; in other words, the funder played an indirect role through the participation of the co-authors.

If the funding organization did not play a role in the study design, data collection and analysis, decision to publish, or preparation of the manuscript and only provided financial support in the form of authors' salaries and/or research materials, please review your statements relating to the author contributions, and ensure you have specifically and accurately indicated the role(s) that these authors had in your study in the Author Contributions section of the online submission form. Please make any necessary amendments directly within this section of the online submission form. 

Please also update your Funding Statement to include the following statement: “The funder provided support in the form of salaries for authors [insert relevant initials], but did not have any additional role in the study design, data collection and analysis, decision to publish, or preparation of the manuscript. The specific roles of these authors are articulated in the ‘author contributions’ section.”

If the funding organization did have an additional role, please state and explain that role within your Funding Statement.

Please also provide an updated Competing Interests Statement declaring this commercial affiliation along with any other relevant declarations relating to employment, consultancy, patents, products in development, or marketed products, etc. 

RESPONSE: Cerner was not a “funder” in the sense that funds were made available to the project, which is why we didn’t include them initially. This has been added and that box has checked. We took this information out of the “acknowledgement section” per formatting requirements. 

A funding statement is now in the comments section of the submission system: 

Cerner provided in-kind support for the study by providing access to data and through the work of Cerner co-authors. Cerner co-authors would have received their regular salary during the course of their work on this project, but this was not linked directly to this manuscript

The competing interests statement is now in the comments section of the submission system, and author contributions are clearly defined in the manuscript:

RESPONSE: The updated financial statement provided outlines this information now in more detail: 

Cerner Corporation-employed co-authors (QH, RT, SP) received salary from Cerner corporation during the time they participated on this research project. Cerner Corp. provided in-kind support for the study through their effort and through providing access to data. Cerner Corp. co-authors were a part of the study team and had roles in creating the dataset and data acquisition (RT, SP), data analysis (QH), the concept and design of the study, data interpretation, and critical revision of the manuscript (QH, RT, SP). EP and AMN receive fees for research consulting to Cerner Corporation outside of the present work. EP and AMN were not compensated for their work on this manuscript. 

RESPONSE: Thank you for the opportunity to clarify. We have added this information to the cover letter:

Data from the Cerner Real World Data COVID dataset are available on request to Cerner corporation Kendra.Stillwell@Cerner.com. Per the user agreements between Cerner and the contributing health systems, all research requests must be approved by the governance council which consists of representatives from both Cerner and contributing health systems. Upon approval, researchers can access the de-identified datasets on the Cerner platform; individual patient data (even de-identified data) are never released outside of the analytic environment to protect patient confidentiality based on the agreements in place between Cerner and participating sites. 

This has also been clarified in the manuscript. 

Per the user agreements between Cerner and participating health systems, individual patient data cannot be shared externally but approved researchers can access the data for individual studies.

RESPONSE: The supplement has been edited to include a figure captain for all captions. 

Reviewers' comments:

RESPONSE: Thank you to the reviewers for the thoughtful and thorough reviews, which have strengthened our manuscript. 

Reviewer's Responses to Questions

Comments to the Author

1. Is the manuscript technically sound, and do the data support the conclusions?

Reviewer #1: Yes

Reviewer #2: Yes

2. Has the statistical analysis been performed appropriately and rigorously? 

Reviewer #1: Yes

Reviewer #2: I Don't Know

3. Have the authors made all data underlying the findings in their manuscript fully available?

Reviewer #1: No

Reviewer #2: Yes

RESPONSE: See above for edits on the data sharing statements

4. Is the manuscript presented in an intelligible fashion and written in standard English?

Reviewer #1: Yes

Reviewer #2: Yes

5. Review Comments to the Author

Reviewer #1: Reviewer comment for PLOS ONE

Date of review: 14 April 2021

Manuscript Number: PONE-D-21-07038

Article Type: Research Article

Full Title: The Impact of Race and Ethnicity on Outcomes in 19,584 adults Hospitalized with COVID-19

Suggested title: The Impact of Race and Ethnicity on Outcomes in 19,584 adults Hospitalized with COVID-19 in the United States

Corresponding Author: Ann Marie Navar, UT Southwestern: The University of Texas Southwestern Medical Center, Dallas, UNITED STATES

Reviewer comment

Summary of the research and overall impression

This manuscript identified whether race and ethnicity impact on risk of death among those hospitalized for COVID-19 in the US. Generally, the paper presented well. The introduction is short and required to add more data about the burden of the disease and background situation in the hospitals or States included in the study. Methods session presents how the analysis was carried out. Results are comprehensive, however, a few editions are suggested.

BTW, I don’t see the line number in the manuscript in PDF version and it is difficult to provide the direction of the comments.

Discussion on specific area of improvement

Abstract

Well-written and clearly presented abstract.

1. Results - line 3: “[22.7% vs 20.8%, unadjusted 1.12 (95% CI 1.02-. I think it should be unadjusted OR 1.12.

RESPONSE: This has been corrected – thank you 

2. The paper was said “in adult”, please add the mean (or median) age of participants in the results.

RESPONSE: This has been added as requested

Through August 2020, 19,584 patients with median age 52 years were hospitalized with COVID-19, including n=4,215 (21.5%) Black and n=5,761 (29.4%) Hispanic persons

Introduction

3. Introduction is too short; it is incomplete information. Suggest to add more data on COVID-19 pandemic outbreak prevalence of (burden) across the US, differences in States, proportion of hospitalized patient and mortality rates, etc. (burden of disease)

RESPONSE: We added data on the burden of disease in COVID-19 to the first paragraph of the intro

The COVID-19 pandemic caused approximately 375,000 deaths in the United States in 2020, and was either the cause of death or contributing cause of death for 11.3% of all deaths in the United States.[ ] Likely due to COVID-19, the age-adjusted mortality increased in the United States by 15.9% in 2020.[ ] Among patients hospitalized for COVID-19, the mortality rate has generally decreased over time, but has been shown to vary across hospitals, with one study showing a 50% variability in risk-standardized event rates of mortality or referral to hospice among COVID-19 patients in the first six months of the pandemic.[ ]

4. Suggest to include other studies findings on race and ethnicity diversion on disease burden and mortality in other infectious diseases and any literature on COVID-19 in other countries. No data is the data to present. (gap in knowledge)

RESPONSE: As this study is focused on the US we did not include race/ethnicity data on COVID-19 from other countries. 

5. Suggest to add how race and ethnicity can influence for diagnosis and treatment for preventing life-threatening situation or reduce the mortality in certain diseases, more appropriately with infectious diseases or COVID-19. (gap in knowledge)

RESPONSE: We added the following to the introduction

Accurate information regarding the impact of race and ethnicity on outcomes is important to help physicians identify higher risk patients when admitted, to identify potential biological mechanisms affecting prognosis, and to target public health interventions appropriately. Most importantly, differences in outcomes by race and ethnicity may serve as an indicator of systemic biases in the healthcare system, with differential treatment leading to differential outcomes.

6. Lastly, why this study is important to conduct (justification) and how this study can provide information or input to the health system.

RESPONSE: See response to the above comment which we feel highlights the importance of understanding variability by race

Materials and methods

7. Pg. 10, 3rd paragraph, 1st line: “ICD-10 diagnosis codes” – please spell this out and add reference.

RESPONSE: This has been spelled out. We noted an error in that we also used MEDCIN and SNOMED codes, this has been clarified in the manuscript. We don’t have a specific reference for ICD-10.

8. Pg. 10, 3rd paragraph, 3rd line: “Cerner EHR” – please spell this out. Add reference for using cut off point for obesity.

RESPONSE: EHR has been fully spelled out. This is a universally used cutoff for BMI for obesity. 

Results

9. Pg. 12, 1st paragraph: to add text about age (said “younger”, but please specify), sex and geographic distribution of cases

RESPONSE: We added the numbers into the text for this section as requested.

Overall, 51.0% (n=9994) of our sample was white while 21.5% were recorded as Black (n=4215). Table 1 shows differences between Black and white adults admitted with COVID-19. Black adults were younger (median age 59 vs 62 years), less likely to be male (47.3% vs 53.0%), less likely to be Hispanic (1.9% vs 36.3%), had higher BMIs (median BMI 30.7 vs 38.9), and different distribution of insurance coverage (p<0.001 for all). Black patients also had higher rates of diabetes, hypertension, coronary artery disease, heart failure, chronic kidney disease, and end stage renal disease, and lower rates of COPD and asthma (see Table 1).

10. Pg. 12, the title is not matched with the text, it is talking about “demographic characteristics and comorbidity of the cases”, not “race and ethnicity”

RESPONSE: On page 12 we are discussing characteristics of patients by race and ethnicity. We discuss differences in comorbidities and demographics of cases by race and ethnicity, which is why this appears in this section.

11. Again, the paragraph does not refer to any table. Please add reference table for the data presented in the text.

RESPONSE: We have clarified this- 

Of the 19,584 patients included, n=4050 (20.7%) died during the hospital stay (Table 2). Table 2 shows characteristics of adults overall and stratified by in-hospital mortality. 

12. Tables: the data (numbers) is suggested to right-aligned, rather than centered.

RESPONSE: We will defer to the editors on the journal preference regarding formatting and choice of left, right, vs center aligned. 

13. Pg. 18, table 3: Please add odds ratios (OR) and CI in the tables. You may use * for p-values that are significant (eg. *p<0.001, #p<0.01)

RESPONSE: We have added these ORs to this table.

Discussion and conclusions

14. Pg. 19, 1st paragraph, line 1: please use the exact sample size, do not use “nearly 20,000 patients” and use “52 health systems”.

RESPONSE: This has been corrected as follows:

In a nationwide, EHR-based database of 19,584 patients from 52 health systems across the United States

15. Pg. 20, 2nd paragraph: line 1-3: please add reference.

This intro sentence sets up the paragraph where we provide specific references.

Recommendation

I would like to propose the minor revision for this manuscript.

Please add the ethical approval number from Duke University.

RESPONSE: This has been added (Pro00105396)

Thank you to the reviewer for the thoughtful and thorough review. 

Reviewer #2: The investigators conducted a retrospective cohort study to evaluate, among patients hospitalized for COVID-19, the differences in race and ethnicity and their association with death.

The manuscript is presented clearly and concisely. The following comments may help in improving the manuscript.

General

1. Persons/patients, consider using one uniformly.

RESPONSE: Except when discussing national data (and referring specifically to populations, not patients), we have changed “persons” to “patients” throughout

2. Mortality/death consider using one uniformly.

RESPONSE: We have changed “death” to “mortality” throughout 

3. clinical comorbidities/comorbidities consider using one uniformly.

RESPONSE: We have simplified and use “comorbidities” throughout

Methods

1. “..we utilized any available encounter within 3 years of and including the analysis hospitalization”. Consider revising this text for clarity.

RESPONSE: This has been clarified

To determine race and ethnicity, we evaluated for the presence of an indicator of race within 3 years of and including the analysis hospitalization.

2. “those discharged on hospice”. Should it be“to” instead of “on”?

RESPONSE: Thank you- this is corrected

Results

1. Consider reporting, % of participants for whom data on race and ethnicity were missing.

RESPONSE: This has been added to the first paragraph

Race data was missing for n=1,017 (5.19%) of patients overall.

2. “Compared with White adults, black adults had similar rates of myocardial infarction, but higher rates of myocardial infarction, stroke, ventricular tachycardia, and pulmonary embolism.” In this sentence, should the text “but higher rates of myocardial infarction” be omitted?

RESPONSE: Thank you- this was an error in the text and has been corrected. 

Compared with White adults, Black adults had similar rates of myocardial infarction, but higher rates of myocardial infarction, stroke, ventricular tachycardia, and pulmonary embolism. 

Tables

1. Expand abbreviations in footnotes (e.g. ESRD)

RESPONSE: Thank you- these have been added 

2. There is no data in Table 1 on complications, and survival, consider revising the Table title

RESPONSE: Thank you for this- we have corrected the title

Table 1: Characteristics of white, Black, Hispanic, and non-Hispanic patients hospitalized with COVID-19 

Discussion

1. Outcome (mortality) data for a significant proportion of participants (about 31%) was not available. Consider mentioning this as a limitation and its likely implications.

RESPONSE: This statement is based on our note that 6.1% were still hospitalized, 1.6% went to hospice, 16.6% were transferred, and discharge status was unknown for 6.5%. We now break this down by race as well in the supplement, and discuss this in the limitations. The overall difference between Black and White patients who were transferred was 3.3%, which is unlikely to impact our findings. Similarly, while there was a higher rate of unknown discharge status in in Black / African American patients, survival would have to have been substantially different in this population to impact findings. 

If transfers were variable by race, and survival was different among those who were transferred vs remained hospitalized, this may have impacted our findings, which may only apply to those patients who are not transferred.

Conclusion

1. Consider revising to exclude redundant text and for clarity.

RESPONSE: We have made revisions throughout the conclusion to help tighten up the messaging as requested.

6. PLOS authors have the option to publish the peer review history of their article (what does this mean?). If published, this will include your full peer review and any attached files.

Do you want your identity to be public for this peer review? For information about this choice, including consent withdrawal, please see our Privacy Policy.

Reviewer #1: Yes: Poe Poe Aung

Reviewer #2: No

---

## [Decision Letter · Decision Letter 1]

5 Jul 2021

The Impact of Race and Ethnicity on Outcomes in 19,584 adults Hospitalized with COVID-19

PONE-D-21-07038R1

Dear Dr. Navar,

We’re pleased to inform you that your manuscript has been judged scientifically suitable for publication and will be formally accepted for publication once it meets all outstanding technical requirements.

Kind regards,

Stanislaw Stepkowski

Academic Editor

PLOS ONE

Additional Editor Comments (optional):

Reviewers' comments:

Reviewer's Responses to Questions

**Comments to the Author**

1. If the authors have adequately addressed your comments raised in a previous round of review and you feel that this manuscript is now acceptable for publication, you may indicate that here to bypass the “Comments to the Author” section, enter your conflict of interest statement in the “Confidential to Editor” section, and submit your "Accept" recommendation.

Reviewer #1: All comments have been addressed

Reviewer #2: All comments have been addressed

2. Is the manuscript technically sound, and do the data support the conclusions?

Reviewer #1: Yes

Reviewer #2: Yes

3. Has the statistical analysis been performed appropriately and rigorously? 

Reviewer #1: Yes

Reviewer #2: Yes

4. Have the authors made all data underlying the findings in their manuscript fully available?

Reviewer #1: Yes

Reviewer #2: Yes

5. Is the manuscript presented in an intelligible fashion and written in standard English?

Reviewer #1: Yes

Reviewer #2: Yes

6. Review Comments to the Author

Reviewer #1: Reviewer comment for PLOS ONE

Date of review: 23 June 2021

Manuscript Number: PONE-D-21-07038_R1

Article Type: Research Article

Full Title: The Impact of Race and Ethnicity on Outcomes in 19,584 adults Hospitalized with COVID-19

Corresponding Author: Ann Marie Navar, UT Southwestern: The University of Texas Southwestern Medical Center, Dallas, UNITED STATES

Reviewer comment

Summary of the research and overall impression

The manuscript was revised according to the reviewer’s comment, improved and ready for publication after revising the title by adding “in the United States”.

Discussion on specific area of improvement

Suggested title: The Impact of Race and Ethnicity on Outcomes in 19,584 adults Hospitalized with COVID-19 in the United States [to add the name of the country at the end of the title]

Recommendation

I would like to accept this manuscript.

Reviewer #2: The authors provided point-by-point response to the reviewers comments and have improved the manuscript.

7. PLOS authors have the option to publish the peer review history of their article (what does this mean?). If published, this will include your full peer review and any attached files.

Reviewer #1: **Yes: **Poe Poe Aung

Reviewer #2: No

---

## [Editor Report · Acceptance letter]

9 Jul 2021

PONE-D-21-07038R1 

The Impact of Race and Ethnicity on Outcomes in 19,584 adults Hospitalized with COVID-19 

Dear Dr. Navar:

I'm pleased to inform you that your manuscript has been deemed suitable for publication in PLOS ONE. Congratulations! Your manuscript is now with our production department. 

Kind regards, 

on behalf of

Dr. Stanislaw Stepkowski 

Academic Editor

PLOS ONE